Decolonisation; global mental health; community engagement; indigenous

**Corresponding author:**
Peter Beresford;
Email: pberes@essex.ac.uk

# Decolonising global mental health: The role of Mad Studies

Peter Beresford[1,2] and Diana Rose[3]

[1]School of Health Science, University of East Anglia, Norwich, UK; [2]Shaping Our Lives, London, UK and [3]CASS and Sociology, Australian National University, Canberra, ACT, Australia

## Abstract

In recent years, there has been a growing and high-profile movement for 'global mental health'. This has been framed in 'psych system' terms and had a particular focus on what has come to be called the 'Global South' or 'low and middle-income countries'. However, an emerging 'Mad Studies' new social movement has also developed as a key challenge to such globalising pressures. This development, however, has itself both being impeded by some of the disempowering foundations of a global mental health approach, as well as coming in for criticism for itself perpetuating some of the same problems as the latter. At the same time, we are also beginning to see it and related concepts like the UNCRPD being given new life and meaning by Global South activists as well as Global North activists. Given such contradictions and complexities, the aim of this paper is to offer an analysis and explore ways forward consistent with decolonizing global mental health and addressing madness and distress more helpfully globally, through a Mad Studies lens.

## Impact statement

We have written this paper in the hope of provoking debate in the general field known as global mental health. The paper puts users/survivors/people with psychosocial disabilities front and centre of thinking about how we should be treated in the Global North and the Global South. Should the first export its ministrations to the second? Many say 'no' but do so from the standpoint of professionals. By drawing on the emerging terrain of Mad Studies we propose a way for the voices of experience to be heard and listened to. But this intervention is not just at the level of debate; Mad Studies is also a praxis and we hope to spark a dialogue about action in the field of 'mental health' in the very diverse and unequal worlds we inhabit.

## Mental health globalisation: Solution or problem

The current move to globalise mental health can be understood as both a trend and a self-conscious development. Whilst colonial countries have psychiatrised their colonies for two centuries, this has mainly been in the replication of institutions. The current move to global mental health is more widespread and co-ordinated. It has been understood to mean the rapid expansion of psychiatric interpretations and responses to human mental distress and difficulty. These have generally been framed in medicalised, individualistic terms, primarily seeing the problem or pathology in the individual, their family or, less often, the wider community.

The first major modern exporting of western psychiatry followed the collapse of the Soviet Union. This tended to be wholesale and undifferentiated between eastern bloc nations, centred on US-dominated big pharma. This resulted in a system which was still heavily institutionalised, generally of poor quality and underfunded (Petrea and Haggenburg, 2014). The researcher China Mills has offered a definitive critique of the much broader Western 'movement for global mental health' which followed, highlighting the overdue need for its decolonization. The call of the World Health Organization (WHO) and the Movement for Global Mental Health was to 'scale up' access to psychological and psychiatric treatments globally, particularly within the Global South (Mills, 2014). While this has been offered as a positive, Mills has raised three fundamental and enduring questions in relation to the 'globalisation of mental health'. These are first, whether the call for equality in global access to psychiatry is a helpful one. Second, whether everyone should have the 'right' to a 'psychotropic citizenship' and third, whether mental health can, or should, be global and appropriately conceived of as a concept with global application. She raises major doubts in all three cases and calls for the decolonisation of mental health.

Mills relies heavily on Fanon and his theory of 'colonising the mind' (Fanon, 1967). However, we cannot approach these questions solely through the work of those from Western societies, however, grounded their claims to follow Southern writers. Importantly, writers have approached decolonialising mental health from within the Global South itself calling for the resurgence of African or South

American epistemologies as the basis for understanding distress or, indeed, for founding new disciplines. (Caroll, 2011). Jessica Horn takes apart the Western idea of 'Post Traumatic Stress Disorder' using African feminist principles including the idea of a collective 'self' (Horn, 2020). She sees 'trauma' as political and a result of structural violence. Specific traumas – which are the focus of the Western diagnosis – are relocated holistically. So, rape is not 'trauma' isolated from the person's life world. It can lead to stigma so severe that the woman cannot feed her family and community as she is rendered economically inactive. Healing is collective and involves not just talk but music and dance and the 'therapist'/'client' relation is levelled in terms of power as all live in conflict-ridden, violent environments and the workers are devalued because of their association with the clients. But these latter are not 'burdens' leading to 'burnout' as in a Western framing; the women have assets which evoke what Horn calls 'vicarious resilience'. The core of this argument is necessarily diluted as we are writing in English as was Horn but her research used Kiswahili, a commonly spoken language in Eastern Congo.

Another example is the work of First Nation Canadian scholar Joseph Gone who has issued a 'provocation' to community psychology to historicize and de-centre its subjects thus calling for a new epistemic space (Gone, 2016). His particular interest is in the 'residential schools' where First Nation Canadian children were stripped of their language, rituals and very ways of being, a situation he conceives as 'intergenerational trauma' (Gone, 2013). He is alive to the nuances of healing when 'hybrid' methods are used because elements taken from the West may be alien to indigenous people's positions.

Finally, decolonial thinking has been applied to research methodology (Keikelame and Swartz, 2019). Keikelame and Swartz make heavy use of the South African idea of 'Ubuntu' which encapsulates the deep interdependency of persons although it is difficult to translate. This is the foundation of 'respect for culture', one of their principles, but that aside the paper reads as a very elaborated example of community participatory research. Of course that may be no bad thing since this form of research praxis arose in the 'Global South', namely in Brazil (Freire, 1996). It is a moot point if Freire's followers in the West do him justice.

None of the above writers are 'survivors' although Horn has claims to be 'one of us'. Activism and knowledge-making by people with psychosocial disabilities in the Global South are complex. We will deal with this in discussing the CRPD. This is not because everything revolves around this Convention, far from it, but it allows us to highlight some of the variety and the differences between survivors in the two regions and thus grapple with the problem of whether what we are proposing is in fact the reverse of decolonial.

It is important to mention at this point that the 'globalisation' of psychiatry denoted by this development is only one expression of the growing reach of the psych 'sciences'. As we have seen from the official manual of the field, which provides for the 'classification of mental disorders', *The Diagnostic and Statistical Manual of Mental Disorders* (latest version) DSM-5-TR (2022), this is also reflected in the growing range of diagnostic categories that have developed; their increasing application in relation to a widening range of social and political problems internationally, and their ever-expanding clientele, from institutionalised older people and sex offenders to dissident school students and trans people (American Psychiatric Association, 2022; Cacciatore and Frances, 2022).

## A problematic alliance

While its reach is still qualified and complex, the dominant global politics in the twenty-first century has been neo-liberal; that is to say based on globalised free market-driven economics with reduced expenditure on supportive welfare services (Beresford, 2016). High-profile concerns have been expressed about this ideological approach and its consequences, evidenced most notably in the work of Wilkinson and Pickett. They argue that inequality in societies, which is particularly associated with free market ideology and politics, is damaging for health, including mental health. (Wilkinson and Pickett, 2009; Pickett and Wilkinson, 2019). The broader connections between 'mental health problems' and political systems have now been well rehearsed by social epidemiologists, although they tend to conceive social determinants of health as 'variables'. Yet psych understandings continue to privilege individualising explanations. Significantly *social* psychiatry which came closest in the discipline to developing more social understandings has been in retreat in recent years internationally, even as psych approaches have extended their authority (Blazer, 2005). A powerful informal alliance has developed between them and prevailing neoliberal politics, glued together by their mutual privileging of individualising, pharma-based responses to distress. An antidote to this is given briefly by Rochelle Burgess and colleagues who call for social interventions in global mental health (Burgess et al., 2020). Their suggestions are redolent of social psychiatry but do not go as far as Horn's and others' 'structural violence'.

## The questioning of psychiatry

The globalisation of mental health is presented as a positive by its proponents, in terms of accessing disadvantaged people's and nations to the benefits of the psych system which originated and expanded in western industrial and post-industrial societies, from which they had previously been deprived. We are told that the plan will help countries achieve UN Sustainable Development Goal target 3.4 – by 2030, reduce by one-third premature deaths from noncommunicable diseases through prevention and treatment, and promote mental health and well-being (cited in Da Costa Paes). However, this seems more likely to be achieved if the impoverishing imperatives of global neoliberalism, especially as they affect the Global South, were to be slowed down. This expression of neo-colonialism is being extended to countries that have both been subjected to colonialism and escaped it in the past. The economic, as well as other, aspects of the UN Sustainable Development Goals can be seen as Eurocentric and thus will have the effect of disadvantaging low and middle-income countries, at all levels, rather than benefitting them.

Furthermore, as Tanya Talaga has written, from Canada to Norway, Brazil, Australia and the United States, the indigenous experience in nations resulting from this colonial legacy is marked by violent separations; of families, individuals, from the spiritual and the land. She reports the resulting intergenerational trauma, rise of mental distress and youth suicide on a global scale (Talaga, 2020; see also Gone, 2013, 2016).

The irony is not lost that as psychiatry has sought to go global and export itself to the poor and middle-income countries of the South, it has also come under increasing question in the global North. This challenge came first from western radical and critical practitioners and intellectuals – like Szasz, Goffman, Scheff, Laing, Cooper, Basaglia and Foucault. Protected to some extent by their professional status and developing their own agendas, they condemned psychiatry as oppressive, abusive, damaging, discriminatory, stigmatising and poorly evidenced (Beresford, 2022). It did not, though, occur to these critics to engage their clients in their

thinking. But if this anti-psychiatry movement could be accused of being narrowly based, this could not be said of its successor. This, the movement, of people directly experiencing madness and distress and/or themselves on the receiving end of mental health policy and provision, emerged from the 1970s. It explicitly identified with other 'new social movements' (NSMs) like the women's, Black civil rights and LBGTQ+, peace and environmental movements, based on identity, experience and attention to social context and power.

## The survivor movement: A NSM

Characteristics associated with these movements include:

- Having a broader focus than traditional labour/economics-focused social struggles;
- Rights-based and committed to social and cultural change, anti-discrimination, equality and social justice;
- Recognising that inequalities extend beyond traditionally understood differences in political power based on socio-economic status;
- Making the connection between direct experience, structural issues and making change;
- Prioritising people being able to speak and act for themselves, self-advocating and self-organising individually and especially collectively;
- Raising issues which transcend national boundaries;
- Highlighting issues of intersectionality, recognising the complexity and interactions of multiple identities and the intersectionality that characterises multiply marginalised bodies when engaged with psychiatry (Fanon, 1967; Touraine, 1981; Oliver, 1996, p. 157; Kendall, 2005).

It is the connections that NSMs make between who we are, our lives, relations and broader ideology and formal politics that have been for many of their advocates, their defining feature. This is highlighted by their stress on the ground-breaking rallying call of the second wave women's movement from the late 1960s, 'the personal is political', which has particular relevance in the present discussion as long as the 'personal' is not confined to the individual (Hanisch, 2006). More generally NSMs have enabled us to recognise the value of difference and diversity and the importance of treating them with equality, rather than perpetuating division.

NSMs are international movements even though they may face different issues locally, make different progress and need to connect with experience in different parts of the world, not least Global South Countries. Most recently this has been highlighted by the #MeToo and Black Lives Matter movements (O'Dowd and Hagan, 2020; Tadros and Edwards, 2021). We will describe some of the differences presently.

One of the early, founding UK survivor-led organisations was Survivors speak Out. In 1987 it agreed a 'Charter of (15) Needs and Demands'. These spelt out an agenda that was common among activist mental health service users at the time and which still has powerful resonance. This includes the valuing of survivors' first-hand views and experience, support for survivor-led and non-medicalised services especially for crisis support, an end to discrimination, the resourcing of self-advocacy, access to medical records, legal protection, an end to ECT and psycho-surgery and independent monitoring of drug use in the system and its consequences. In a subsequent statement, SSO condemned the medical model, demanded treatment choice, adequate welfare benefits, support for self-advocacy, more say in policy and 'the right to be

valued for what we are and …might become…not for what we were or were thought to be'. Participants unanimously opposed compulsory treatment in the community (SSO, 1987) The UK history developed by the Survivors' History Group, a survivor-led initiative, also highlights the longstanding efforts, with qualified success, in the survivor movement to address issues of equality in relation to gender, race, disability and sexuality as well as early mentions of the international nature of the movement (Survivor History Group, n.d.).

## A global movement

While the survivor movement may have had origins in the Global North, there has always been activism in the Global South too. And, indeed, it would be wrong to homogenise the Global South – TIA (Towards Inclusion Asia), for example, is not internally of a piece and other priorities supervene in other regions. In particular, there is little psychiatry in the Global South, few mental hospitals and few psychiatrists and this is what the 'Gap' is trying to remedy. Therefore resistance must be different as we showed above. Tellingly, the organisation 'The Pan-African Network of Users and Survivors of Psychiatry' changed its name in the 2010s to the 'Pan-African Network of Persons with Psychosocial Disabilities'. Its reasoning was basically: we do not have much psychiatry and we do not want it. This invites us to approach our project with care. Mad Studies grew out of high-income countries with developed psy services and will need to be adapted to be helpful in the Global South. However, there may be more points of commonality than are apparent at first sight.

In 2018, for example, the UK government hosted a Global Ministerial Mental Health Summit. Its policy track record was poor in regard to survivors and it failed to involve service users in the event, to invite any survivor-led organisations and only one service user from the Global South citing, of all things, cost! Two open letters of opposition were published from service users and allies. One came predominantly from user-led organisations in the North. The other was supported by survivors, their organisations and allies in the South, from more than 20 countries including Peru, India, the Asia Pacific region, Argentina, Columbia, Singapore, Brazil, Australia, the Philippines, Hong Kong, Shanghai, Kenya and South Korea, highlighting the concerted and truly global nature of this opposition (NSUN, 2018; Pring, 2018).

## GMH under challenge

The globalisation of mental health has been presented by its advocates as a progressive strategy, accessing the Global South to the same benefits of the psych system as operate in the advantaged Global North nations. Human Rights Watch has reported people with mental health conditions living in chains and confined in small spaces in Asia, Africa, Europe, the Middle East and the Americas (Human Rights Watch, 2020). The cruelty of such responses is not in question, but it ignores what have been called the big problems of 'slow' or 'structural violence' associated with the psychiatric system, arising from practices of restraint, seclusion, forced treatment and neglect, constraint and unaccountability (Mills, 2014, p. 106; Daley et al., 2019; Reel, 2019; Voronka, 2019; Horn, 2020) Given the continuing highlighting of structural racism in advanced systems of psychiatry, like that highlighted by the Black UK activist Colin King and the failure to address Afrocentric models of distress, it is difficult to be reassured (King and Jeynes, 2021; King, 2022). This is

close to Caroll's argument too, situated as it is in Africa (Caroll, 2011)

The NSUN-sponsored global response to the 2018 Global Ministerial Mental Health Summit as well as the international one highlighted in microcosm some of the key themes in global survivor activism, including the need to:

- Involve services users, theirs and other civil organisations;
- Address the discrimination black and minoritised communities and migrants from ex-colonial countries and the Global South diaspora face;
- Call a halt to unevidenced western anti-stigma programmes;
- Support local, inclusive innovations in the South to address social and structural determinants of health rather than take over leadership (Davar, 2016; NSUN, 2018).

### Challenging orthodoxies

One of the themes of survivor movements has been their apparent reluctance to replace one orthodoxy with another. For most western mental health service users, the prevailing orthodoxy is that imposed by the medicalised individual model of 'mental illness' and its associated diagnostic categories and models of treatment. For many of us, this seems an incredibly powerful interpretation of our emotions, feelings and experience, which can be difficult to disregard (Beresford, 2010). Rejection of such conceptualization is even likely to result in our rationality and relation with reality being called into question.

The disabled people's movement has developed the 'social model of disability' which draws a distinction between the individual's actual or perceived impairment and the discriminatory and disabling societal reaction this often prompts. This has been widely adopted in public policy internationally. However, research suggests that many mental health service users/survivors are reluctant to adopt such a model for themselves. Some fear that if they do, they will be seen as in denial and this is interpreted as part of their disorder. Others regard their inclusion in the category of disability as groundless and further pathologizing and reject the social model of disability for themselves as they do not see themselves as having an impairment. Many do not wish to have a yet another stigmatising label attached to them as 'disabled'. They do seem however to be widely critical of a medical model of mental illness and much more supportive of a social approach to understanding distress which takes account of its social construction and the multiplicity of reasons for it (Beresford et al., 2009, 2016). However, as we shall be seeing, this lack of a clear, agreed philosophical core makes it vulnerable to subversion by the psych system. However, Mad Studies may have a different answer to this. In addition, the antipathy to 'disability' does not seem to be the case in the Global South where the term 'persons with psychosocial disabilities' is becoming increasingly used.

### Problems with global mental health

If such a diverse response should emerge in one country, then it seems extremely unlikely that in the Global South with its massive political, cultural, social and historical diversity or that the varied perspectives and shared rights of people experiencing madness and distress would be adequately served by a uniform system like western psychiatry. This diversity is expressed in the intersectional differences visited on marginalised people in the South, when engaged with psychiatry, which can include violence as we have seen. It also undercuts the generalisations typical of the UN Sustainable Development Goals, which are not only general, but also Western at that.

Although the mental health model encourages individualised explanations, it cannot itself readily be understood in isolation. It is historically closely bound up with the western age of enlightenment and the values associated with that in relation to 'science', rationality, knowledge production, economic and colonial expansion. Critics of Global Mental Health (GMH) conclude that mental health practices are 'situated within systems of power and colonial hegemony' and draw connections between GMH and neocolonialism, including:

- European colonial hegemonic beliefs;
- Western domination, systemic oppression and discrimination;
- The erasure and appropriation of traditional (indigenous) values

and they argue for the decolonizing of mental health practice (Millner et al., 2021). This needs to be a process which takes account of the pervasive influence of coloniality and challenges it. A Chilean study highlights the failure of approaches which are trapped in the colonial system of power/knowledge and life/being rather than seeking to transcend it by drawing on indigenous knowledge and concepts (Jara and Pisani, 2020).

### Decolonising knowledge: A shared concern

Here we can see a helpful meeting point with survivor approaches to knowledge development which seek to value what has come to be called 'lived experience' and experiential knowledge (Rose, 2022a, 2022b). This has encountered an inherent problem though in seeking to challenge western traditional valuing of positivist approaches to research, with their emphasis on 'objectivity', 'neutrality' and 'distance'. These are still strong in the psych system and have distorted our understandings of what counts as knowledge. So, if you have direct experience of a problem, like poverty, distress or indeed colonisation, where such research values apply, you can expect to be granted less credibility and your knowledge seen as less reliable because you are 'too close to the problem' – it affects you and you cannot claim to be neutral, objective and distant from it. Thus, you can expect to be seen as an inferior knower and your knowledge less reliable. This means effectively that if you have experience of discrimination and oppression you can expect routinely to face *further* discrimination and be *further* marginalised and devalued (Beresford, 2003). This has come to be known as epistemic injustice and inequality (Fricker, 2007) Significantly, this is a situation that faces people both as mental health service users/survivors and as the subjects of colonisation. For example, Dabashi has written a book with the title 'Can Non-Europeans Think?' (Dabashi, 2015).

### Mad Studies

This brings us, by contrast to Mad Studies, which may be seen as a movement that could be well placed to challenge GMH, because it constitutes such a clear break from it and its colonialist history. It speaks to the observation of the African-American feminist Audre Lorde that 'the master's tools will never dismantle the master's house', offering the possibility of an alternative.

As we have argued, the lack of a unifying philosophy in the survivors' movement, has left it exposed to co-option and incorporation by the psych system. As the South African academic Femi

Eromosele suggests there are some in the survivors' movement seeking reform, others revolution. However, a more recent offshoot of the survivors' movement, *Mad Studies* presents a much less ambiguous challenge to the psych system and global mental health. As Eremosele says, Mad studies has increasingly come to be identified with the latter perspective, 'seeking a total change in society's definition of madness' (Eremosele, 2020, p. 177).

It originated with the survivor movement in Canada but has since spread throughout Europe and may have traction in the Global South, brought to prominence by the founding text, *Mad Matters* (LeFrancois et al., 2013). Mad studies is a movement, a discipline and a form of activism, thus it is a praxis. It can be seen as the first survivor-led movement which has sought to develop strong philosophical and theoretical principles. The clear principles that seem to demarcate and reflect its process and purpose are that:

- It rejects a bio-medical model of our 'mental well-being';
- It is based on a rights, social and holistic, rather than individualised understandings and approaches;
- It values and gives priority to survivors' lived experience and first-person experiential knowledge;
- It is survivor-led but not limited to survivors;
- It builds on collectivist approaches to understanding, organising and making change;
- It seeks to build broader alliances beyond 'mental health';
- It is clearly ideologically and theoretically based;
- While it is survivor-led it is open to all to be involved who accept its principles.

Thus, Mad Studies is less a 'model' than a set of conditions and principles that allow for differences and this can be seen as constructive. Most probably, both are needed and the emphases will vary. In this context, It has been argued that decolonial thinking and practice must focus on the local and locally-determined action and this by very high-profile scholars and political leaders (Martin, 1987; Perch et al., 2012). We would rather argue for a balance between this and means to learn from each other and forge alliances *on equal terms*, which would involve some general principles, both between those positioned as in distress in different regions and with other marginalised groups.

Mad Studies has nonetheless from its early days come in for significant criticism, registered both by survivors as well as others, which has focused particularly on:

- Worries about its potentially narrow base restricted to an elite, needing to address difference with equality;
- Its over-academicisation – over-reliant on space in and the values of the Western academy;
- Its use of a defining term Mad, which is still strongly devalued and contested in many societies (Rose, 2022a, 2022b).

Critical friends emphasise the importance of Mad studies, being seen as a 'developing project', recognising such 'tensions' (Spandler and Poursanidou, 2019) and of it seeking (and its potential), to create alternative counter-cultures of critical inquiry, support and solidarity (Sweeney, 2016).

We are now gaining Mad Studies insights into the effects of colonisation and ways forward to decolonise from many parts of the majority world. There is a sizable and growing literature based in and exploring the Global South from a Mad Studies perspective. The recent *International Handbook of Mad Studies* includes contributions from and about developments in India, Latin America and Africa (Sandborn, 2022). Texts have emerged, for example,

from Ghana, India and Kenya, with more in progress. (Nabbali, 2013; Davar, 2016; Sharma, 2022).

## Mad Studies: An alternative with emancipatory potential

Mad Studies has encouraged its proponents to move beyond a psychiatry-anti-psychiatry binary, based primarily on the Global North experience, to pay more critical attention to interventions in the Global South and their own role in this, both as local activists and allied survivors. Thus the survivor researcher Jasna Russo has recently written of the importance of us as survivor activists ensuring that we rid ourselves of any continuing psychiatric influences in our own work, to depsychiatrise it, to avoid enforcing 'the very phenomenon we seek to expose disrupt' (Russo, 2022). There are parallels here of strong relevance for decolonisation, with the way we internalise the model of mental health and can learn from the work of people like Franz Fanon and Paulo Freire and concepts of whiteness and conscientization how to challenge the identities imposed on us (Mills, 2014; Da Costa Paes, 2021). Mad Studies helps us to see the relations of madness with society, and its maddening effects, as well as the interconnections of 'mental health' with colonisation. The significance of this for Mad Studies in the South is evident.

While they should not be overstated, there are parallels between those victimised by colonisation and those who seek collectively to challenge their mental health status, policy and practice. Historically subjects of mental health policy and practice have had:

- Their rights restricted and subjected to enforced 'treatment';
- Been segregated and congregated in separate institutions;
- Been conceived of as inferior, pathological and deviant;
- Racialised and subjected to institutionalised racism;
- Had their political and other rights reduced and removed.

While not suggesting that mental health service users are the equivalent of colonised populations, there are echoes that can make for understanding and solidarity and there is a critical space for intersectionality, as we have explained. This and their commitment to inclusion and to the decolonialisation of mental health suggests Mad Studies may offer both a helpful lens and force for change, enhance understanding and increase challenges.

We have been seeing activists and their organisations in the South, resisting and moving beyond the narrow confines of psychiatry and psychiatric thinking to form theoretical and practical alliances with other movements; of poor, homeless and criminalised people, disabled and indigenous people, to reinforce and expand each others' understandings and create new collaborations and solidarities. Their search for and development of new approaches to mental well-being contrasts strongly with the political retreat from mental health and other social policy in countries like the UK which continue to be strongly subject to neoliberal ideology. This also reflects the global experience of Covid-19 where death tolls have tended to correlate with the kind of neoliberal politics associated with GMH and been among the highest in these marginalised groups, both North and South.

Mad studies offers a route to decolonisation (of GMH) consistent with decolonising aims and values. Thus:

- It is collective.
- Ideologically committed, but culturally and philosophically open.
- Participatory rather than directive.

- Committed to inclusion and the valuing of experiential knowledges and diversity.

This is one of the strengths of Mad Studies. It is not necessarily tied to one approach or having one expression, as we have argued. It appears to embrace a variety of different ideas and philosophies consistent both with its own principles and those of decolonisation (Beresford and Russo, 2022) Thus it side-steps the critique of not having a 'model' applicable everywhere. It serves both in decolonising mental health and offering alternative ways of developing appropriate arrangements for people's mental well-being, as local survivor activists:

- Connect with local values and beliefs rather than seeking to override them;
- Offer opportunities for collective involvement, rather than individualised external imposed intervention;
- Build grassroots organisations that provide the spaces for personal empowerment – rethinking ourselves – as a first step to joining together to direct our own collective action. Indeed, this is not a linear 'person to the collective' for the collective also shapes the person.

However, as should be evident by now, Mad Studies originated in high-income, Eurocentric societies and cannot be adopted wholesale in the very different contexts of the Global South. For this, we need to look for points of connection between the approach and the knowledge-making and practices that have been and are developing in the Global South. For instance, the Western (and highly psychologised) notion of Recovery has been critiqued by Mad studies scholars but there is also a sizeable literature critiquing its individualism and egocentrism from the Global South (Topor et al., 2011; Bayetti et al., 2016; Kaiser et al., 2020; Rose, 2022a, 2022b). Sumeet Jain has explicitly linked such critiques to the movement for people with psychosocial disabilities in India (Jain, 2016). This literature is not service-user led; tellingly one searches in vain for such concerns on the part of people with distress in the Global South.

In this context, one of the approaches adopted by both activists and policymakers in the Global South is that offered by the United Nations Convention on the Rights of People with Disabilities (UNCRPD). This draws on the philosophy of independent living developed by the international disabled people's movement, based on the idea of ensuring that disabled people are enabled to live on as equal terms as non-disabled people. The notion of independent living should be seen historically – as a shift from and contrast to Institutionalisation – rather than as individualised.

There are differences of emphasis in the responses of people with distress to the CRPD as between the Global North and the Global South. For instance, the controversial General Comment to Article 12, focussing on the right to legal capacity, has been discussed in the Global North largely in relation to involuntary psychiatric commitment. In the parts of the Global South the range is much wider – the right to marry, to sign contracts, to receive an education because although involuntary commitment exists psychiatric hospitals are sparse (and awful, modelled on the Victorian asylum). Again, Towards Inclusion Asia highlights Article 19 – the right to social inclusion. This is taken up too by Australian aboriginal writer Scott Avery, who is Deaf with a cochlear implant (Avery, 2018). The CRPD is not homogenous in terms of rights – it includes individual, practical and social rights. These differences of emphasis we refer to partly mirrors these social and cultural distinctions. Any rapprochement between Mad Studies and the movements of people with psychosocial disabilities will entail Mad Studies learning from these differences. It cannot be a one-way street.

And so critiquing individualistic western rights-based discourse, is to recognise the value and importance of indigenous concepts that can help in the understanding of madness and distress. This includes the African concept of Ubuntu, which we have encountered already, meaning belief in a universal bond of sharing that connects all humanity. As the African researcher Eromosele notes:

> Decolonisation means different things to different people. (It) must go beyond a resort to mere indigenisation – the uncritical adoption of certain models…just because they emanate from the continent. (Decolonisation) must focus not so much on the regional provenance (of ideas like Ubuntu) but on their ideological import and usefulness for the immediate context (Eromosele, 2022, pp. 336–337).

The People's Charter for an Eco-Social World (2022) linked with the first People's Global Summit sponsored by the UN and international welfare organisations in the summer of 2022 similarly highlighted Ubuntu, global rights and Buen Vivir – an indigenous social movement from South America that describes a way of life and a form of development that sees social, cultural, environmental and economic issues working together and in balance (The People's Charter for an Eco-Social World, 2022).

Mad Studies, which featured in this Summit, developed along the lines articulated by its global supporters, offers the prospect of helping us all better understand and improve our mental well-being, as well as reconceiving the 'global mental health' paradigm that continues to do damage to both those who perpetuate it and those damaged by its colonising impact. Just as it is important to build equal and inclusive alliances between different NSMs, if they are to achieve greater traction, so it is essential to establish more equal and inclusive dialogue between activist movements in the Global South and north which explore different understandings and realities. Resource issues inhibit this, but in the new normality of a post-Covid world, virtual communication offers new possibilities as well as reinforcing old barriers. We offer these thoughts as a starting point. Much remains to be done and dialogue is crucial.

**Open peer review.** To view the open peer review materials for this article, please visit http://doi.org/10.1017/gmh.2023.21.

**Author contribution.** P.B. was solely responsible for the first draft of this manuscript. After serving as a peer reviewer, D.R. was invited to act as co-author and significantly contributed to the revised version of the manuscript.

**Financial support.** This research received no specific grant from any funding agency, commercial or not-for-profit sectors.

**Competing interest.** The authors declare none.

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
