## [Reviewer Report]

Dear Editor of Global Mental Health,

Please find attached my submission for your consideration for the journal.

Thank you very much for your time.

Best wishes,

Peter

---

## [Reviewer Report]

This is a clearly written piece that offers an alternative perspective on global mental health from the dominant ones (including those on decolonizing gmh). Although I don’t agree with all that is in it, that is not a reason to criticize it in terms of publishing it. The historical references are overall accurate, scholarly, and contribute depth, in fact more than usual for papers on gmh. The author acknowledges the problems and challenges faced by a movement rooted in this language, at the same time as conveying the reasons for hopefulness about it. In short, I think it’s important for voices from this perspective to be more widely heard, and to become part of the discussion of gmh. This piece provides an excellent basis for furthering that process.

---

## [Reviewer Report]

I need to begin by acknowledging that I declared to the editors of this journal that I had a conflict of interest with respect to the paper. That is, I know the author both professionally and personally. However, the Editorial Board had such difficult finding reviewers in what they called a ‘small’ and ‘niche’ field that they asked me to set this aside but at the same time be transparent about it. This I have tried to do and I will flag where there are differences of opinion and where I am suggesting development of arguments, which are just that, suggestions.

The paper is an ambitious attempt to situate Mad Studies in the contentious field of Global Mental Health. And how it might be decolonised. Mad Studies is held to have an advantage over previous critiques of ‘psy’ because it is based on the actual experience of distress and responses to it. In the West these are dominated by ‘psy’ and in the Global South they are not. The project of GMH is to ‘scale-up’ Western psychiatry into the ‘gap’ that exists in the Global South. This point is well-made. However, my first suggestion concerns reference to the ‘anti-psychiatrists’. They are not just ‘protected by their professionalism’ but it never occurred to them to involve us in their projects. Neither did they touch on colonialism. So, there are other critiques of psychiatry more directly linked to colonisation which would be worth mentioning. China Mills is referenced but one thinks also of Summerfield and Fernando. This is simply by way of situating the argument about experience.

Much is made of the Global Ministerial Mental Health Summit held in the UK in 2018 which was indeed an important event and the tensions are well-described. People from the Global South were underrepresented though and this had a lot to do with the denial of funds to bring such representatives to London. It is said often but funding is important. The WHO Quality Rights programme is incredibly well-funded; even more so the Mental Health Gap (MhGAP). I know the author knows this but it could be emphasised.

I have two main suggestions. First, in claiming that the movement against GMH is ‘Global’, the author invokes the World Network of Users and Survivors of Psychiatry. This in my view is not the best example. It has an office in Denmark but no telephone number and they do not answer letters. More importantly, it has a ‘line’ (we have no impairments all is oppression) and surely we do not want to replace one globalising system with another. Of course, the author may disagree with that analysis. But they do emphasise the diversity of the Global South and in place write of the importance of the local. An example of a regional organisation is TIA (Towards Inclusion Asia)which of course is not the same all over Asia but it is an important example not least because it emphasises different parts of the CRPD than do some writers from the Global North. Even the contentious General Comment on the right to legal capacity (Article 12) is differently situated. In the Global North discussions are confined to the Emergency Room whereas the settings are much broader in the Global South. This is probably too much detail for a paper but I would caution against assuming that even the CRPD is interpreted and emphasised in a homogenous way. I appreciate the author seeks to draw diverse positions together to posit a ‘global movement’. But that is quite possible whilst acknowledging differences explicitly.

My second point is equally just a suggestion. I think it is important to emphasise that ‘psy’ is not just psychiatry. Movements like ‘Recovery’, spearheaded by psychologists, have been criticised for their individualism, by writers such as Bayetti and Jain. Their critique is that in more collectivist societies the ideas of ‘autonomy’ and ‘self-determination’ do not align with cultural practices. This whole area – probably too detailed for the paper – of ego-centric, socio-centric, eco-centric and spirit-centric societies (ideal types of course) – has been argued compellingly by Joseph Gone who writes on intergenerational trauma in First Nation Canadian societies. Gone issues a ‘provocation’ to community psychology to move away from ahistorical, individualistic framings of the person. So we have again a concrete example of how ‘individualism’ can be superseded. All this is quite consistent with the arguments in the manuscript. Another example is ‘task shifting’. The much bemoaned lack of psychiatrists and other ‘psy’ professisons in the GS is to be corrected by handing over ‘tasks’ to local people. But of course they do not just hand over tasks, a whole narrative is being disseminated as they do this.

These thoughts and some others were sparked by reading this manuscript. I definitely think the manuscript should be published but some more concrete examples would strengthen it. They do not have to be my examples of course. I shall desist from suggesting others. But, to conclude, we need papers like this to show the potential reach of Mad Studies and to enter it into new spaces.

**Following this review, the peer reviewer became a co-author on the manuscript and took no further part in the peer review process.**

---

## [Reviewer Report]

My congratulations to the authors for such an interesting piece, which focuses on a series of very important challenges facing the global mental health field. While I think that the dialogue that this paper seeks to advance is one that is worthy of publication, the work in its current form cannot be published. In addition to smaller points, which I will outline below, the major flaw feels to me, to be conceptual, and stems from a potentially narrow understanding of decolonial thought and praxis.

First, it feels like a contradiction to call for the decolonialisation of a field that is rooted in the imposition of western frames, through the introduction of yet another prism of yet another western frame. MAD studies, while an incredibly important and radical field and space for activism and knowledge generation around the violences of mental health - originates in high income cultures and settings. This cannot be the foundation for a decolonial process, if it is not rooted in knowledge systems and paradigms that are truly ‘southern’. If the author disagrees with this, then much more effort needs to be put into defending this choice, with relevance decolonial scholarship itself - and southern perspectives on mental health knowledge/praxis. The current paragraph on the ‘global’ nature of the survivor does not problematise the notion of who has access to participating in these movements, and who is excluded, or what forms of knowledge beyond embodied rejection of diagnostic categories are picked up or excluded by such a paradigm. Beyond this, it doesn’t deal with the fact that at the heart of nearly every African and latin epistemological/ontological/cosmological frame - is the core of a collective in place of the individual. I don’t feel that the MAD studies frame does enough to meaningfully create space for this - but remain open to being convinced if the author is able to elaborate on these points.

Other smaller points are as follows (listed by page number of the document, rather than the PDF):

Introduction/abstract

pg.1 The global south is a term that predates the global movement - something about the way this is written needs revisiting - as it seems to suggest that these to phrases are chronologically linked

Mental health globalisation: solution of problem

pg 1 There has been a much longer historical expansion of western psychiatry to the global south, which dates back to the 1800’s with Colonialism. Please see the work of following scholars:

Parle, Julie. “Mental Illness, Psychiatry, and the South African State, 1800s to 2018.” In Oxford Research Encyclopedia of African History. 2019.Parle, Julie. “Mental Illness, Psychiatry, and the South African State, 1800s to 2018.” In Oxford Research Encyclopedia of African History. 2019.

pg 3

While the work of Mills is important, it is not appropriate for you to pin the argument for decolonialisation of a field in the arguments and frameworks entirely of a western scholar. Mills acknowledges that she is in the tradition of Fanon - who is a mandatory reference to consider/include. More recent work has been completed by Jessica Horn on this, she is an African feminist scholar and it would strengthen your claims to engage explicitly with this body of work.

There is also a growing argument for social interventions in global mental health, that takes root in very similar principles to social psychiatry. It is worth articulating this nascent approach to global mental health as a starting place for redressing critique.

Please see the following:

https://pubmed.ncbi.nlm.nih.gov/31653556/

Mad studies

Pg 9

don’t feel as though drawing on a western knowledge system - no matter how critical, allows us to fully grapple with the framings of African/indigenous ways of being.

See the work of Black sociologists in search of African world views - such as Karajana Carrol, https://journals.sagepub.com/doi/abs/10.1177/0896920512452022

Mad Studies - emancipatory potential

Pg 11 What you fail to articulate here is the multiple intersections of difference and oppression that historically marginalised bodies endure, when they are also identified and constructed within a psychiatric frame. This is an important consideration within Fanon’s work.

pg 13 - Why can’t this just be achieved, through locally driven activism? embodied and shaped by local principles and actors who drive for change in ways that they articulate? what is the explicit value in the insertion of this framework, to ‘decolonise’ when it is from the north? Who is this for? are you calling for academics to pick up the banner of Mad studies? or everyday citizens? is this supposed to be a way for western actors who seek to do GMH to organise their practise? or everyday people mobilising? if it is the former, then maybe I could see that. But if it is the latter, I disagree, whole heartedly for a call to drive people in any direction, that is not self-determined direction. And that is in principle what decolonial principles are about (See the writings of Thomas Sankara, Amal Cabral)

pg 14

The comment on how activists make use of the UN principles is for me, the exact place where this argument starts to be lost. ‘Independent living’ as a concept, and the individualised paradigms of psychiatry as a whole (as noted earlier in this work) are at odds with many Southern modes of being and knowing. How can you meaningfully make connections between these western ideological frames - such as those embodied by the UN and rights based frameworks, and these principles?? Again, the work of Connor linked to in earlier comments will be helpful to think through these things here.

Overall comments - There are a few missed words and typos that need addressing are throughout the document. But these are minor and not problematic on the whole.

The use of sub-headings isn’t always helpful - there may be too many of them, and I wondered if some sections of the essay were brought together, it may make the argument stronger i.e put all the critiques of GMH in one place, rather than interspacing them with discussions of social movements